# Membrane Adaptations and Cellular Responses of *Sulfolobus acidocaldarius* to the Allylamine Terbinafine

**DOI:** 10.3390/ijms24087328

**Published:** 2023-04-15

**Authors:** Alka Rao, Niels A. W. de Kok, Arnold J. M. Driessen

**Affiliations:** 1Department of Molecular Microbiology, Groningen Biomolecular Science and Biotechnology Institute, University of Groningen, 9747 AG Groningen, The Netherlands; 2Department of Chemical and Pharmaceutical Biology, Groningen Research Institute of Pharmacy, University of Groningen, Antonius Deusinglaan 1, 9713 AV Groningen, The Netherlands

**Keywords:** GDGT, archaea, membrane, *Sulfolobus*, isoprenoids, caldariellaquinone, terbinafine

## Abstract

Cellular membranes are essential for compartmentalization, maintenance of permeability, and fluidity in all three domains of life. Archaea belong to the third domain of life and have a distinct phospholipid composition. Membrane lipids of archaea are ether-linked molecules, specifically bilayer-forming dialkyl glycerol diethers (DGDs) and monolayer-forming glycerol dialkyl glycerol tetraethers (GDGTs). The antifungal allylamine terbinafine has been proposed as an inhibitor of GDGT biosynthesis in archaea based on radiolabel incorporation studies. The exact target(s) and mechanism of action of terbinafine in archaea remain elusive. *Sulfolobus acidocaldarius* is a strictly aerobic crenarchaeon thriving in a thermoacidophilic environment, and its membrane is dominated by GDGTs. Here, we comprehensively analyzed the lipidome and transcriptome of *S. acidocaldarius* in the presence of terbinafine. Depletion of GDGTs and the accompanying accumulation of DGDs upon treatment with terbinafine were growth phase-dependent. Additionally, a major shift in the saturation of caldariellaquinones was observed, which resulted in the accumulation of unsaturated molecules. Transcriptomic data indicated that terbinafine has a multitude of effects, including significant differential expression of genes in the respiratory complex, motility, cell envelope, fatty acid metabolism, and GDGT cyclization. Combined, these findings suggest that the response of *S. acidocaldarius* to terbinafine inhibition involves respiratory stress and the differential expression of genes involved in isoprenoid biosynthesis and saturation.

## 1. Introduction

Crenarchaeal membranes are highly dynamic, as their composition is dependent on environmental factors, such as nutrient availability, pH, and temperature. Ether-linked archaeal lipids have been extensively studied because of their unique chemical properties, rendering membranes more robust with reduced proton permeability. In recent years, the lipid biosynthesis pathway in archaea has been well-characterized. It commences with the isoprenoid building blocks, isopentenyl pyrophosphate (IPP) or dimethylallyl pyrophosphate (DMAPP), from the alternate or classical mevalonate pathway [1]. *Sulfolobales* utilize the classic mevalonate pathway (MVA) for this step [2]. IPP and DMAPP undergo sequential condensation reactions with geranylgeranyl pyrophosphate synthase (GGPPS) to form GGPP (Figure 1) [1]. The glycerol-1-phosphate (G1P) backbone is synthesized by G1P dehydrogenase (EgsA) in *Sulfolobus acidocaldarius* (Figure 1) [3]. GGPP is further processed by geranylgeranyl glycerol phosphate (GGGP) synthase (GGGPS) to form GGGP with a glycerol-1-phosphate backbone (Figure 1) [1]. The formation of the diether 2,3-O-geranylgeranylglyceryl diphosphate (DGGGP) is catalyzed by DGGGP synthase (DGGGPS) [1,4]. CDP Archaeol synthase or CarS then activates DGGGP through cytidine triphosphate (CTP) to produce CDP-archaeol [5]. In the next step, the cytidine diphosphate (CDP) head group is replaced with polar head groups such as inositol, glycerol, or ethanolamine [5]. Geranylgeranyl reductase (GGR) is responsible for the hydrogenation of the unsaturated DGGGP to yield a saturated archaeol or 2,3-di-O-phytanyl-sn-glyceryl phosphate or archaetidic acid (AA) [6,7]. The substrate specificity of GGR remains unclear; however, it has been demonstrated that CarS is specific for unsaturated substrates, suggesting that saturation is a downstream process [8]. In archaea, tetraether lipids can be formed by tail-to-tail condensation of two diether lipids [9]. These tetraether lipids, termed glycerol dialkyl glycerol tetraethers (GDGTs), span the membrane. Recently, the enzyme involved in this process was identified, tetraether synthase (Tes), which belongs to the family of radical S-adenosylmethionine (rSAM) and catalyzes reactions through the formation of free radicals [10]. GDGTs can incorporate up to eight cyclopentane rings into their structures. Two GDGT ring synthases, GrsA and GrsB, have been identified in *S. acidocaldarius* (Figure 1) [11]. These proteins belong to the rSAM family. GrsA and GrsB introduce rings at the C-7 or C-3 position in GDGTs, and their expression is regulated by pH or temperature [11,12]. These GDGTs can be modified by the addition of headgroups. The calditol or nonnitol headgroup of GDGTs is synthesized by Cds, another rSAM enzyme that yields glycerol dialkyl nonnitol tetraethers (GDNTs) [13].

The cellular membrane of *S. acidocaldarius* predominantly consists of monolayer-forming GDGTs with a lower abundance of GDNTs [14,15]. Inositol-phosphate dialkyl glycerol diether (IP-DGD) is the most abundant bilayer forming dialkyl glycerol diether lipid (DGD) in the *Sulfolobus* membrane [15]. The membrane composition of *S. acidocaldarius* is modulated by environmental factors, such as growth phase, growth rate, temperature, pH, and starvation [11,15,16,17]. Common adaptations include altering the ratio of DGDs to GDGTs and incorporating cyclopentane rings into tetraether lipids. These adaptations presumably lead to a reduction in the permeability of crenarchaeal membranes at higher temperatures [18]. Various headgroups can be found in GDGT lipids of *Sulfolobus* including inositol-phosphate, monohexose, dihexose, calditol, or nonnitol [15,19]. The calditol headgroup has an adaptive function for survival in acidic environments [13]. Apart from ether-linked lipids, crenarchaeal membranes contain other isoprenoids such as respiratory quinones. The abundance of these quinones is correlated with pH, temperature, and salinity in archaea [20]. Essentially, quinones with short acyl chains are correlated with thermophiles, whereas the presence of long-acyl chains is correlated with salinity [20]. Based on these correlations, quinones have been proposed as putative membrane regulators of archaea [20]. The respiratory quinones of *S. acidocaldarius* include caldariellaquinones (CQs) and sulfoquinones (SQs) [21]. Biochemical studies have recognized CQs as a pool of reducing factors for cytochromes in the electron transport chain [22,23,24]. Notably, the distribution of CQs is restricted to *Sulfolobales,* and their higher mid-redox potential (compared to menaquinones) could be an adaptation to strictly aerobic metabolism [20,21]. The complete biosynthetic pathway for CQs remains to be elucidated; however, GGR from *S. acidocaldarius* is known to accept GGPP as a substrate (Figure 1) [6]. This leads to the formation of phytyl diphosphate, which has been proposed as a precursor for CQs [6].

Radiolabeling assays with the crenarchaeon *Thermoplasma acidophilum* have indicated the accumulation of DGDs upon treatment with terbinafine, a squalene epoxidase inhibitor in fungi [25]. This effect was not observed in *Halobacterium salinarum*, which has a membrane lacking GDGTs [25]. It has been suggested that this compound inhibits tetraether synthase, which is now identified as Tes [25]. The target of terbinafine in the fungal membrane is squalene epoxidase, an FAD-dependent enzyme involved in the ergosterol biosynthesis pathway [26]. Molecular dynamics simulations have indicated that the lipophilic part of terbinafine binds to the central cavity of squalene epoxidase and induces conformational changes that block the substrate from entering the binding site [26]. Since archaea do not synthesize sterols, the target(s) and mechanism of action of terbinafine remain elusive. Additionally, studies with a comprehensive lipidome analysis and the cellular responses of archaea to terbinafine are absent from the literature. In this study, the impact of terbinafine on the lipidome of *S. acidocaldarius* was determined, and transcriptomic analysis was conducted to gain insights into the cellular responses of crenarchaeotes to terbinafine. Furthermore, the high concentrations of terbinafine required for growth inhibition suggest that the inhibitory mechanism might be more complex than originally anticipated in targeting specifically the tetraether synthase. 

## 2. Results

### 2.1. Terbinafine Causes the Accumulation of DGDs and Depletion of DGTs in S. acidocaldarius Membranes

*S. acidocaldarius* MW001 was grown in the presence of various concentrations of terbinafine in Brock medium at 75 °C, pH 3.0 with aeration. Terbinafine inhibited *S. acidocaldarius* growth in a concentration-dependent manner (Figure 2). The lipidome of the MW001 strain was analyzed during the exponential and stationary phases of growth in the absence and presence of terbinafine (Figure 3, Table 1). The accumulation of inositol phosphate dialkyl glycerol diether lipid (IP-DGD, *m*/*z* 893.685 [M-H]^−^) was observed in exponential and stationary phase cells (Figure 3 and Figure 4) [15,27]. IP-DGD was the only diether lipid that showed consistent accumulation across growth phases (Figure 3 and Figure 4). The levels of DGGGP (*m*/*z* 715.507 [M-H]^−^) and archaetidic acid (AA, *m*/*z* 731.632 [M-H]^−^) were slightly elevated during the exponential phase; however, they were depleted during the stationary phase (Figure 3 and Figure 4) [15,27]. The lipidome of *S. acidocaldarius* was examined for GDGTs with various cyclopentane rings (0–8) [27]. A slight decrease in the levels of GDGT-1 to 3 was observed in the exponential phase of growth with terbinafine (Appendix A). Meanwhile, GDGT-1 to 6 showed slightly decreased levels in the stationary phase of growth with terbinafine (Appendix A).

A reduction in levels of all abundant GDGTs including di-hexose inositol phosphate GDGT 0–6 (*m*/*z* 1866.433–1854.339 [M-H]^−^), penta-hexose inositol phosphate GDGT-0 to 6 (*m*/*z* 1542.327–1530.233, [M-H]^−^), di-hexose GDGT—0 to 6 (*m*/*z* 1670.419–1658.293, [M + CHO_2_^−^]^−^), and GDGT- 0 to 6 (*m*/*z* 1346.314–1334.220, [M + CHO_2_^−^]^−^) was observed only in the stationary phase (Figure 5B) [15,27]. Interestingly, this relative reduction in GDGT levels was not observed during the exponential growth phase (Figure 5A).

### 2.2. Terbinafine Interferes with the Respiratory Complex in S. acidocaldarius

To examine the early cellular responses of *S. acidocaldarius* on terbinafine-induced growth inhibition, the transcriptomic response was evaluated after three hours of growth with terbinafine at concentrations of 0.31 mM and 1.22 mM. At 0.31 mM terbinafine, only five genes were upregulated (at *p*-adjusted values < 0.05): *aceA* (isocitrate lyase), *argG-H* (argininosuccinate synthase and lyase), *leu2* (3-isopropylmalate dehydratase large subunit), and saci_2320 (predicted glutamate synthase) (Appendix A) [28]. In *S. solfataricus*, the uptake of glutamate, leucine, and isoleucine from the medium is the highest among all observed amino acids [28]. Simulations indicated that leucine undergoes incomplete degradation to 3-methyl-2-butenoate, which was detected in the culture supernatant [28]. The biosynthesis and degradation of amino acids may be potential targets for lower concentrations of terbinafine.

Meanwhile, 260 genes were differentially expressed with 1.22 mM terbinafine. These genes were mapped to the arCOG database. The density plot (Figure 6A) illustrates the categorization of these genes based on arCOGs. Most of the differentially expressed genes (*p*-adjusted value < 0.05) were mapped to the arCOG pathway ‘Energy production and conversion’, as illustrated by the column graph (Figure 6B). *S. acidocaldarius* is a strict aerobe which conserves energy by a proton driven chemiosmotic gradient [29]. Transcript levels of *sdhA*, *sdhC*, and *nuoD* were elevated, which are part of the succinate dehydrogenase (SDH) and NADH complexes responsible for reducing quinones [22,30] (Figure 7). This respiratory chain has three terminal oxidase complexes: SoxABCDL, SoxEFGHIM, and DoxBCE [31]. The transcript levels of *soxA-C*, *soxG-I*, and *doxB-C* were also elevated (Figure 7). The ATP synthase complex consists of nine subunits: *atpA-I* [23]. Transcript levels of *atpA* and *atpB* were elevated (Figure 7). This was accompanied by depleted transcript levels of proposed acyl-CoA dehydrogenases, elevated levels of acetoacyl-CoA acetyltransferase and enoyl-CoA hydratase (Appendix A). 

### 2.3. A Shift in the Saturation Levels of Caldariellaquinone

Caldariellaquinones (CQ) act as electron and proton carriers in the respiratory chain of thermoacidophiles [29,31]. *Sulfolobus* predominantly synthesizes two classes of benzothiophenones: sulfoquinones and CQs [32]. They are found only in saturated forms of crenarchaeotes [21]. *S. solfataricus* alters its quinone composition depending on temperature, carbon source, and availability of oxygen [32]. Specifically, the total quinone content increases with increasing aeration [32]. Only the saturated species of caldariellaquinone (CQH, *m*/*z*: 630.451 [M]) were detected in the lipid extracts of *S. acidocaldarius* without terbinafine (Figure 8) [21]. Monosaturated (CQ 6:1, *m*/*z* 627.428, [M-H-]^−^), di-saturated (CQ 6:2, *m*/*z* 625.412, [M-H-]^−^), and tri-saturated (CQ 6:3, *m*/*z* 623.396, [M-H-]^−^) species and reduced levels of CQH were observed in the exponential and stationary phase lipid extracts after growth with terbinafine (Figure 8). For CQH, fragments corresponding to the loss of the headgroup were observed, as reported previously (Figure 8C) [21]. The polyunsaturated species, CQ 6:1 and CQ 6:3, showed additional product ions related to the loss of parts of the isoprenoid chain (Figure 8C), which has been reported for other quinones, such as menaquinone (MK) [21].

An MK-specific reductase from *Archaeoglobus fulgidus* was shown to alter the MK saturation profile in *Escherichia coli* [33]. WP_011278262 (saci_1431) in *S. acidocaldarius* is a homolog of this enzyme with 29% sequence identity. The transcript levels of this gene were unaffected. GGPP is a proposed common precursor of isoprenoids such as CQs and membrane lipids [34]. The transcript levels of GGGPS were unaffected (Figure 9), and GGPP could not be detected in the lipid extracts of *S. acidocaldarius*.

### 2.4. Influence of Terbinafine on the Expression of Phospholipid Biosynthesis Genes

RNA sequencing data indicated that terbinafine caused a significant elevation in saci_0240 (*grsB*) transcript levels, which has been identified as a GDGT cyclization enzyme in *S. acidocaldarius* [11] (Figure 9). However, cyclization of GDGTs was not significantly affected. A growth phase-dependent decrease was observed in the levels of GDGT-1 to 6 (Appendix A). This could likely be a response of the membrane to energy availability and not necessarily due to the elevated transcript level of *grsB* [35]. Transcriptomic data also indicated significantly depleted levels of *egsA* (saci_0640), which is responsible for the NAD(P)H- dependent reduction of dihydroxyacetonephosphate (DHAP) to produce a glycerol-1-phosphate backbone for archaeal lipids (Figure 9). Interestingly, terbinafine did not significantly affect the transcript levels of saci_0703 (Figure 9), which was recently identified as tetraether lipid synthase (Tes) [10].

### 2.5. Cell Envelope and Motility

In *S. acidocaldarius*, motility is an ATP–dependent physiological process mediated by the archaellum [36,37]. It is a rotating-type IV pilus consisting of seven proteins (flaBFGHIJX) transcriptionally regulated by *arnR*, *arnR1*, and *arnB* [36,38,39,40]. The *flaB* promoter is induced by environmental stressors such as nitrogen starvation [41]. *ArnR* binds to the *flaB* promoter under nutrient-limitation conditions [40]. Both *flaF* and *flaG* are conserved components of the archaellum, localized in the membrane [36]. Meanwhile, *flaH* and *flaI* are predicted cytoplasmic components. Transcript levels of *flaB*, *flaF*, *flaG*, *flaH, flaI*, and *arnR* were elevated, whereas *flaG* was reduced in the presence of terbinafine (Appendix A). In addition, the transcript levels of *slaA* in the cell envelope were elevated (Appendix A) [42].

## 3. Discussion

The lipid membranes of thermoacidophilic organisms are highly adaptive owing to the extreme environments in which these organisms thrive [11,13,15,35,43,44]. Terbinafine has been proposed as a tetraether lipid biosynthesis inhibitor in archaea based on radiolabeled [2-^14^C]mevanolic acid incorporation studies [25]. However, the concentrations needed for inhibition are substantially higher than those required in eukaryotes, i.e., millimolar versus micromolar, respectively. Therefore, it remains uncertain whether terbinafine is a specific inhibitor of tetraether lipid biosynthesis or whether it has global inhibitory effects on cells. Therefore, the cellular response of *S. acidocaldarius* to this compound was studied. The inhibitory effect of terbinafine on growth and GDGT biosynthesis was consistent with the results of a study in *T. acidophilum* [25]. However, transcriptomic analysis has shown that growth inhibition is accompanied by altered expression of several subunits of the respiratory system. In addition to altering the distribution of GDGTs over IP-DGD, terbinafine inhibition, the precursor for GDGT biosynthesis, also leads to a shift of the CQs to unsaturated species and, to a lesser extent, a reduced level of cyclization of the GDGTs.

Terbinafine inhibits squalene epoxidase in fungi [26]. Squalene epoxidase is involved in the biosynthesis of ergosterols, a membrane regulator responsible for the fluidity and permeability of membranes [45]. In bacteria, hopanoids have similar function [46]. Both sterols and hopanoids can induce ordered phases in lipid membranes [20]. Sterols and hopanoids have not been reported in archaea. Possible candidates for these functions in archaea could be other polyterpenes: carotenoids, polyprenols, quinones, and polyisoprenoids [20,47]. Therefore, it is possible that isoprenoids like the CQs belong to the targets of terbinafine in archaea. The shift towards unsaturated CQs species starts to occur in the exponential phase and continues in the stationary phase. Respiratory quinones in archaea have been proposed to serve as membrane regulators [20]. Hyperthermophiles, in particular, harbor quinones with short acyl chains, such as CQs, which could aid in membrane packing and the improvement of lipid chain order [20]. It is difficult to assess the significance of saturated CQs in *S. acidocaldarius* because little is known about these isoprenoids. Unsaturated CQs could influence the fluidity of the membrane as well as the ordered phases of the membrane. So far, the distribution of CQs is known to be restricted to *Sulfolobales* [20,21,32,48]. The only known archaeal quinone reductase is from *Archaeglobus fulgidus* which solely synthesizes menaquinones (MK) [21,33]. This class of enzymes is paralogous to archaeal GGRs, and their multiplicity in genomes indicates that such enzymes could be responsible for the saturation of other isoprenoids, such as CQs [33]. *S. acidocaldarius* harbors five such uncharacterized paralogs to GGR which are clustered into arCOG00570. Thus, these paralogs could be responsible for altering the saturation profile of the CQs.

The Fad_Rsa_ cluster has predicted acyl-CoA dehydrogenases, 3-hydroxyalkyl-CoA-dehydrogenase, and acetyl-CoA acetyltransferase in *S. acidocaldarius* which are involved in the synthesis of acetoacetyl-CoA, a precursor of the alternate mevalonate pathway in archaea [2,49]. The altered transcript levels of these genes may indicate a potential bottleneck in the synthesis of the isoprenoid building blocks—IPP and DMAPP (Appendix A). However, the transcript levels of most genes involved in the ether lipid biosynthesis pathway remained unaffected by terbinafine inhibition, whereas the precursor GGPP could not be detected in the lipidome (Figure 9).

Our study confirms the growth phase-dependent depletion of GDGTs in the presence of terbinafine in *S. acidocaldarius*, as previously observed for *T. acidophilum* [25]. Significant DGT depletion was observed only in stationary-phase cells, indicating that alteration of the lipidome upon terbinafine inhibition is a slow event. Additionally, the transcript levels of the recently identified tetraether lipid synthase, Tes, were not affected after three hours of growth with terbinafine. Taken together, these findings suggest that depletion of GDGTs may be a secondary effect of terbinafine on the lipidome. IP-DGD is the predominant fully saturated diether in the membrane of *Sulfolobus*, and its amounts are consistent across growth phases [2,17]. Unsaturated archaetidylglycerol (AG) has been proposed as a precursor for tetraether lipids in *T. acidophilum* based on radiolabeling assays [50]. Remarkably, IP-DGD was the only diether lipid that increased in level upon addition of terbinafine (Figure 4). The discovery of Tes has fully not resolved the question as to whether the precursor molecule needs to be saturated or (partially) unsaturated; however, purified Tes protein from *Methanosarcina acetivorans* shows a preference for the fully saturated diether species in a lipid extract [9].

The archaeal membrane maintains metabolic processes through energy-transducing processes such as the transfer of electrons/protons or ATPase-dependent translocation of ions [51]. The altered transcript levels of the Fad_Rsa_ cluster, respiratory chain complex, *grsB,* archaellum, and *slaA* in the cell envelope indicate that terbinafine induces a multitude of responses in *S. acidocaldarius* related to energy metabolism. The synthesis of tetraether lipids and formation of cyclopentane rings are key adaptations that regulate rigidity and homeostasis [2,51,52,53]. Membranes formed with higher levels of IP-DGD, unsaturated CQs, and lower levels of tetraether lipids might be less rigid and have higher basal permeability. The surface protein layer (S-layer) is a major component of archaeal cell walls and is, therefore, critical for survival [42]. SlaA is an impetus for the assembly of the S-layer [42]. An elevation in its transcript levels under the influence of terbinafine could indicate a structural deformity of the cell envelope [54]. This could also explain the terbinafine-induced growth inhibition observed in *S. acidocaldarius*. The presence of unsaturated CQs in the membrane, along with the upregulation of genes involved in the respiratory complex, indicates that the flow of electrons is disrupted. Furthermore, the altered expression of the F1F0-ATPase subunits suggests a possible problem with ATP synthesis in cells. A possible explanation for these membrane adaptations and transcriptome responses to terbinafine is that the organism adjusts to this respiratory stress by redirecting its metabolic energy to increase motility through the archaellum and not by investing its net reducing power in the saturation of CQs or synthesizing complex GDGTs. However, proteomics analysis and assessment of the impact of terbinafine on the respiratory capabilities of the organism may further support this conclusion.

This study describes the cellular response of *S. acidocaldarius* to terbinafine. Terbinafine has a multitude of effects on *S. acidocaldarius*, one of which is the altered DGD: GDGT ratio in the membrane. Considering its effects on the respiratory chain and saturation of CQs, terbinafine appears to be an indirect inhibitor of GDGT biosynthesis. The role of other isoprenoids in archaeal membrane architecture, and their regulatory mechanisms remain elusive. Furthermore, knowledge of quorum sensing in *S. acidocaldarius* is limited to lactonase, and the sensors responsible for membrane adaptations are not known [55]. Therefore, the present studies on archaeal membrane adaptations rely on the correlation of changes in lipid species with the environments inhabited by these organisms. CQs have not been quantified in the *Sulfolobus* membrane so far, making it difficult to speculate on any possible structural role of these isoprenoids. This is the first study to show that an allylamine drug—like terbinafine—affects the saturation profile of respiratory quinone (CQ) in the membrane of *S. acidocaldarius.* This drug causes respiratory stress in organisms, causing a redirection of its metabolism towards energy conservation and motility. This could explain the reduced levels of complex phospholipids, such as GDGTs, in the membrane. 

## 4. Materials and Methods

### 4.1. Strains and Growth Conditions

*S. acidocaldarius* MW001 was grown at 75 °C by shaking at 120 rpm in Brock medium, pH 3 supplemented with 0.1% NZ-amine, 0.2% dextrin, and 10 µg/mL uracil. As indicated, various amounts of a stock solution containing 250 mg/mL terbinafine hydrochloride (Sigma-Aldrich Chemie NV, Zwijndrecht, the Netherlands; final concentrations of 0.31, 0.61, and 1.22 mM which are in the same range as the earlier study with *T. acidophilum* [25]) dissolved in DMSO were added such that its concentration in the medium did not exceed 0.16%. Growth curves were generated from 5 mL cultures (in biological triplicates), and growth was monitored in time at OD_600 nm_ up to 72 h. The initial inoculums were at an OD_600 nm_ of 0.01. 

### 4.2. RNA Isolation, Sequencing and Transcriptome Analysis

For RNA isolation, *S. acidocaldarius* MW001 cultures (50 mL) were grown in duplicate until the exponential phase (OD_600 nm_ of 0.3) and then subjected to terbinafine (0.31 mM and 1.22 mM) or DMSO treatment (0.16%) for 3 h. Cultures were harvested (Allegra X-R benchtop cooled centrifuge, 3000× *g*, 20 min, 4 °C), and cells were processed further to isolate RNA using the TRIzol method [43]. The obtained RNA was purified by ethanol precipitation. Ribosomal RNA depletion, library preparation, quantification, and RNA sequencing were performed by GenomeScan BV, Leiden. Ribosomal RNA was depleted using the NEBNext rRNA depletion kit (Bioke, New England BioLabs Inc., Leiden, the Netherlands) for bacteria. Sequencing was performed using paired-end reads of 150 bp, with a sequencing depth of 20 million reads. The sequenced reads were aligned against the *S. acidocaldarius* DSM639 genome (RefSeq ID: NC_007181.1) using STAR [56]. These reads were mapped and visualized using multiQC for quality control [57]. Differential gene expression analysis was performed using DESeq2 [58]. Adjusted *p*-values (<0.05) were used for data analysis to reduce the false discovery rate (FDR). The results of differential gene expression from DESeq2 were mapped to the archaeal cluster of orthologous genes (arCOGs) and then manually revised [59]. Figures were constructed using standard graphics editing software based on MetaboMAPS [60].

### 4.3. Lipid Extraction and Analyses

*S. acidocaldarius* MW001 cultures were inoculated in 200 mL of Brock medium supplemented with either 0.16% DMSO or 1.22 mM terbinafine in biological triplicates. Cultures were harvested by centrifugation (3000× *g*, 4 °C, 20 min) at the exponential (22 h) and early stationary phases (40 h) of growth. The pellets were washed with fresh Brock medium, centrifuged again, flash-frozen in liquid nitrogen, and freeze-dried at 0.07 mbar, −55 °C for 48 h. Freeze-dried biomass (10 mg) was processed for lipid extraction using a modified acidic Bligh and Dyer method [61] using 0.1 M HCl. Di-oleoylphosphatidylglycerol (DOPG, 5 µg) was added as an internal standard at the beginning of the Bligh and Dyer extraction method. This internal standard was used to normalize the total lipid content of each sample for relative quantification. The chloroform fraction obtained from Bligh and Dyer extraction was evaporated under a N_2_ stream to form a lipid film, re-extracted with chloroform, and dried again [27]. This step was repeated twice using 1:2 chloroform-methanol and methanol [27]. The obtained lipid film was weighed (MS104TS/00, Mettler Toledo BV, Tiel, the Netherlands) and dissolved in methanol (0.25 mg/mL) for UHPLC-MS analysis. The analysis was performed on an Accela1250 UHPLC system (Thermo Fisher Scientific, Breda, the Netherlands) coupled to a Thermo Exactive Orbitrap mass spectrometer (Thermo Fisher Scientific, Breda, the Netherlands) equipped with an ESI ion source in negative mode. *Sulfolobus* lipids were separated using an Acquity UPLC CSH C18 column (2.1 × 5 mm, 1.7 μm) with an eluent flow rate of 300 µL/min at 55°C. A scan range of *m*/*z* 125–2500 was used in the full MS mode. The voltage parameters were 3 kV (spray), −75 V (capillary), −190 V (tube lens), and −46 V (Skimmer). Eluent A was MilliQ water (Millipore Plus System, resistivity 18.2 mΩ.cm): MeCN (40:60), and Eluent B was MilliQ water: MeCN:1-BuOH (0.5:10:90), both containing 5 mM ammonium formate (pH 6.5) [30]. A linear elution gradient was used: 55/45 eluent A/B for 2.5 min, 10/90 eluent A/B from 2.5 to 24.5 min, and returning to 55/45 eluent A/B for 25–33 min. Spectra were acquired in parallel-reaction-monitoring mode for the inclusion list masses (calculated for negative mode) at low resolution, followed by MS/MS fragmentation of the targeted precursor ions with a resolution of 17,500. The collision energy potential was set at 30 or 60 V. Lipid species were identified by their molecular weight, retention time, and fragmentation analysis using an in-house calculated in-silico lipid database for archaea [30]. Peak areas were integrated and calculated with the genesis algorithm using Xcalibur software (Thermo Fisher) with a 5 ppm mass window for peak detection. The obtained fragmentation spectra were compared and validated with published studies [6]. Extracted ion chromatograms were visualized using xcms and MSNbase packages in R [62,63]. Statistical significance was determined using the Student’s T-test in R 4.2.1 [64].

## Figures and Tables

**Figure 1 ijms-24-07328-f001:**
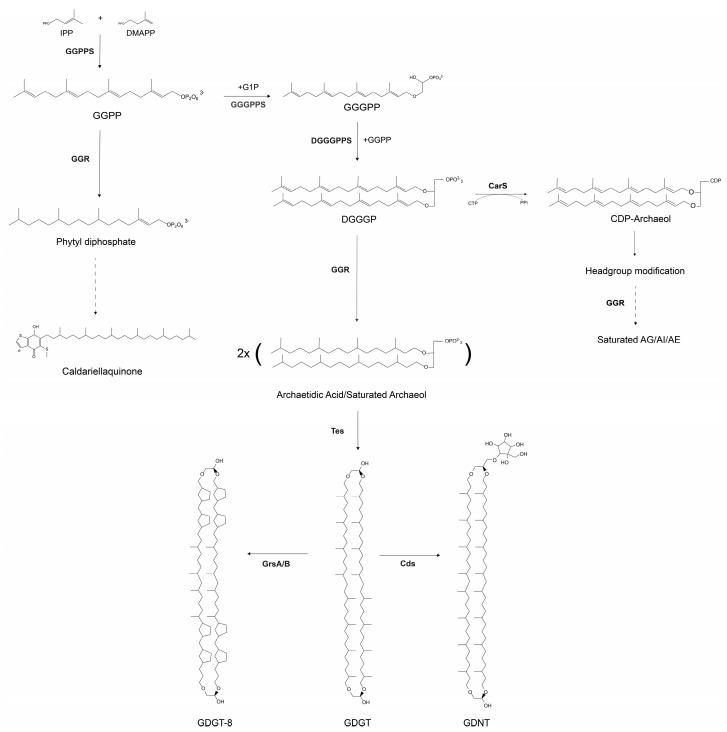
Schematic overview of the archaeal lipid biosynthesis pathway. The dashed arrows indicate a multi-step pathway that has not yet been characterized. AG-archaetidylglycerol, AE-archaetidylethanoamine, AI-archaetidylinositol.

**Figure 2 ijms-24-07328-f002:**
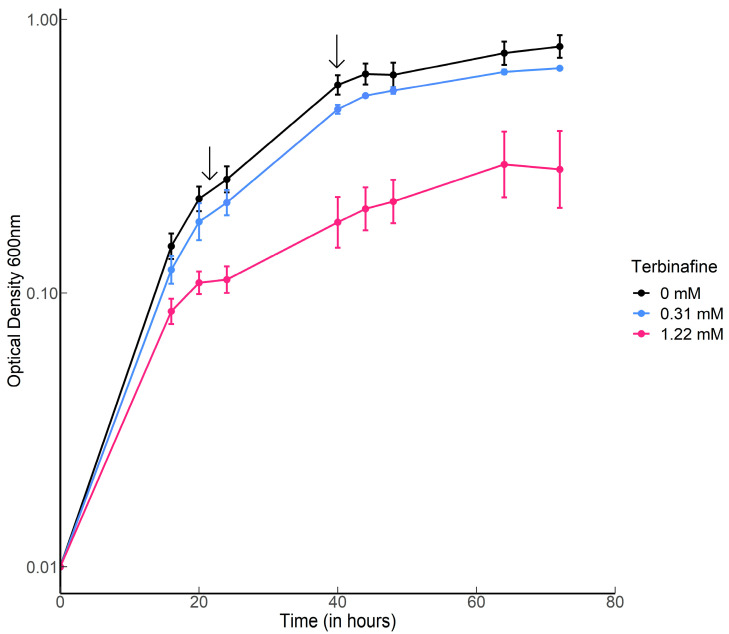
Growth curves of *S. acidocaldarius* with and without terbinafine: Error bars represent the standard error of the mean. Black arrows indicate the time points for harvesting the cells for lipid extraction.

**Figure 3 ijms-24-07328-f003:**
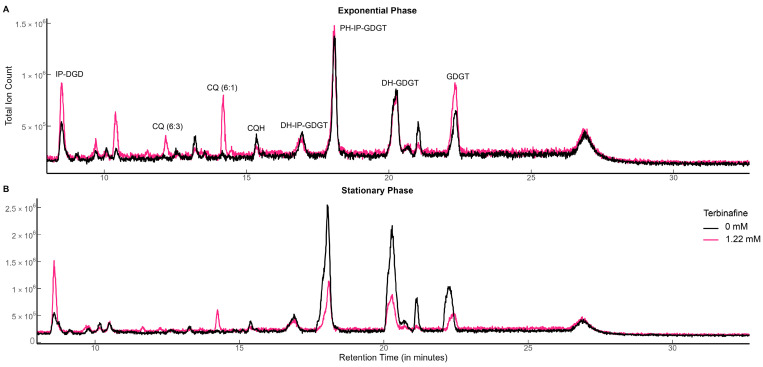
Total ion chromatograms of *S. acidocaldarius* lipid extracts with and without terbinafine: (**A**) exponential phase and (**B**) stationary phase. TIC was normalized to the amount of lipids using the internal standard DOPG.

**Figure 4 ijms-24-07328-f004:**
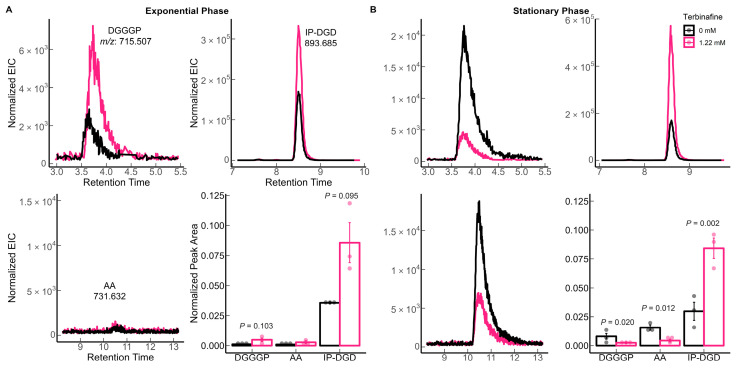
Effect of terbinafine on the bilayer-forming membrane lipids (DGDs) of *S. acidocaldarius*: Extracted ion chromatograms and relative quantification of DGGGP (digeranylgeranylglyceryl phosphate), AA (archaetidic acid), IP-DGD (inositol phosphate dialkyl glycerol diether) in (**A**) exponential phase and (**B**) stationary phase of growth. Error bars represent the standard error of the mean and *p*-values are indicated for each lipid species. The dots represent biological replicates. The theoretical *m*/*z* [M-H]^–^ values are listed in the figure. The extracted ion chromatograms (EICs) were normalized to the amount of lipids using the internal standard DOPG.

**Figure 5 ijms-24-07328-f005:**
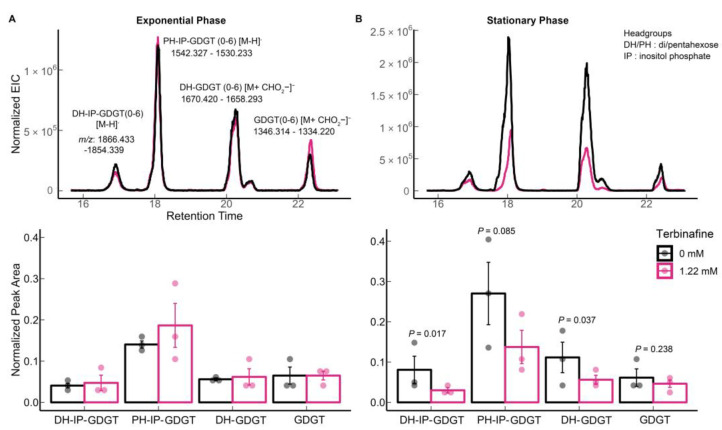
Effect of terbinafine on monolayer forming membrane lipids (GDGTs) of *S. acidocaldarius*: Extracted ion chromatograms and relative quantification of tetraether lipids in (**A**) exponential phase and (**B**) stationary phase of growth. Error bars represent the standard error of the mean, and *p*-values are indicated for each lipid species. The numbers in parentheses indicate the number of cyclopentane rings. The dots represent biological replicates. The theoretical *m*/*z* values are shown in the figure. The EICs values were normalized to the amount of lipids using the internal standard DOPG.

**Figure 6 ijms-24-07328-f006:**
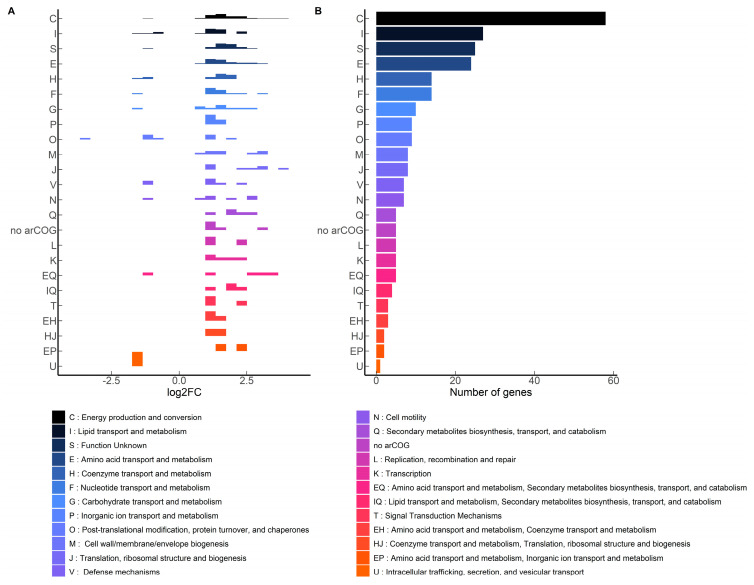
arCOG mapping of significantly affected MW001 genes with terbinafine (*p*-adjusted < 0.05): (**A**) Density plot illustrating log2 fold change distribution of genes, (**B**) Tally of genes mapping to the arCOG categories.

**Figure 7 ijms-24-07328-f007:**
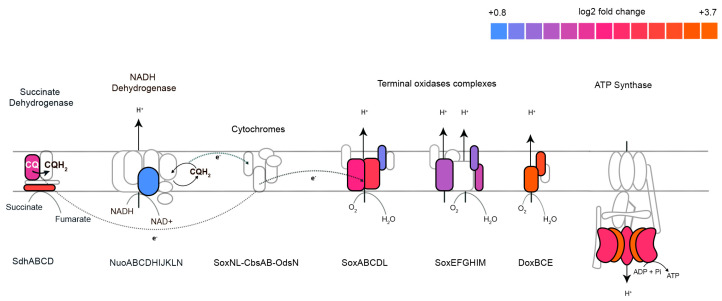
Effect of terbinafine on the transcription of the *S. acidocaldarius* respiratory complex: Colors represent log2 fold change values. No color indicates significantly unaffected transcript levels. All log2 fold-change values are *p*-adjusted < 0.05. CQ refers to caldariellaquinone.

**Figure 8 ijms-24-07328-f008:**
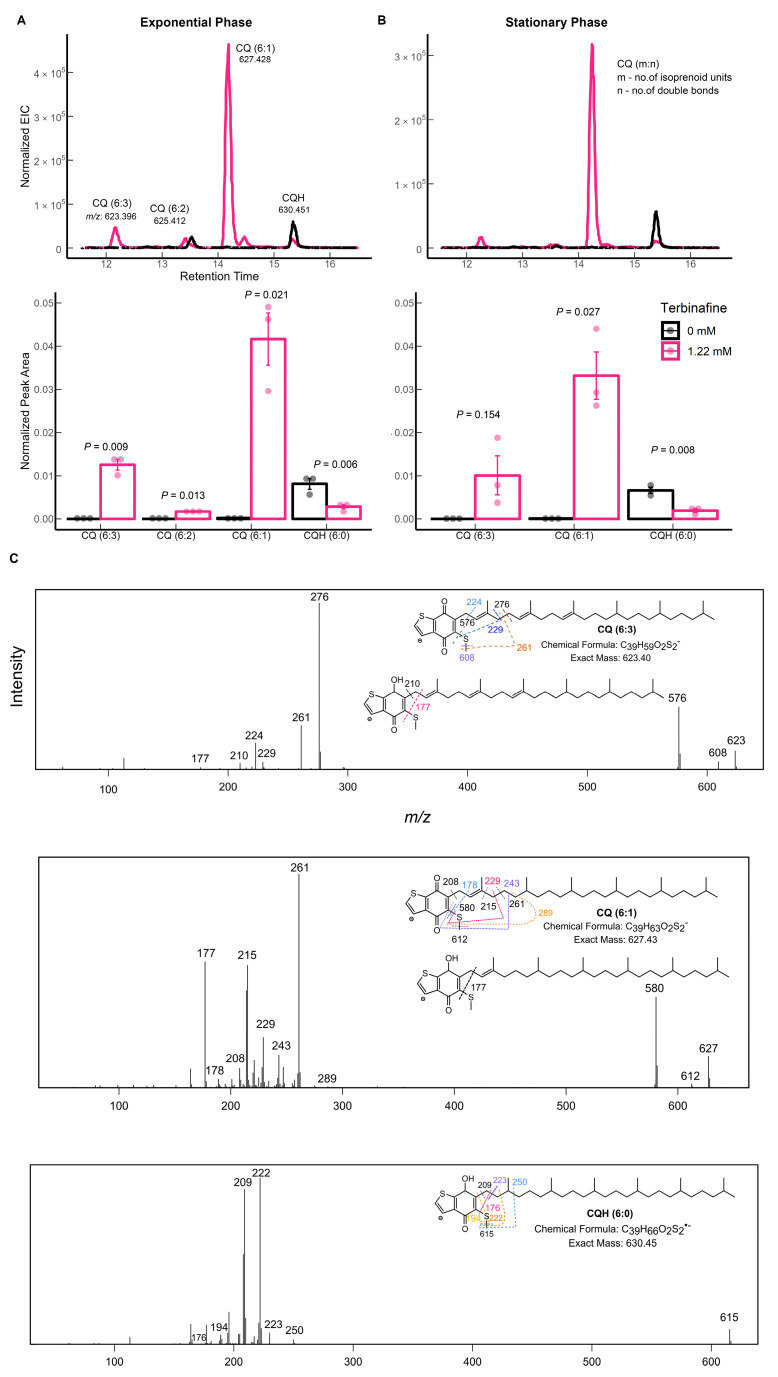
Altered saturation of caldariellaquinone (CQ) under the influence of terbinafine: (**A**) extracted ion chromatogram (top) and relative quantification of peak areas (bottom) in the exponential phase of growth in *S. acidocaldarius* and (**B**) stationary phase. Error bars represent the standard error of the mean, and *p*-values are indicated for each lipid species. The dots represent biological replicates. (**C**) Fragmentation patterns of the observed CQ species. The EICs values were normalized to the amount of lipids using the internal standard DOPG.

**Figure 9 ijms-24-07328-f009:**
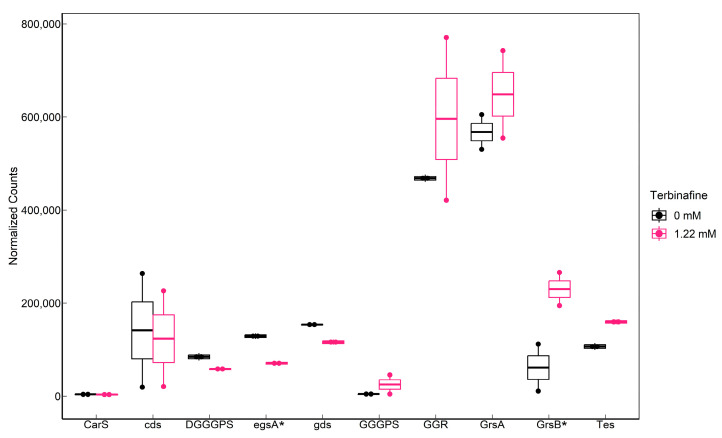
Transcriptomic response of lipid biosynthesis genes in *S. acidocaldarius* to terbinafine: Significantly affected genes (*p*-adjusted < 0.05) are indicated by an asterisk (*) on the x-axis. Counts were normalized for sequencing depth using the mean ratio method in DESeq2. *cds*: saci_1498 (calditol synthase), *carS*: Saci_0897 (CDP-archaeol synthase), DGGGPS: Saci_1565 (Digeranylgeranylglycerol phosphate synthase), *egsA*: Saci_0640 (Glycerol-1-phosphate dehydrogenase), *gds*: Saci_0092 (geranylgeranylpyrophosphate synthase), GGGPS: Saci_0728 (geranylgeranylglycerolphosphate synthase), GGR: Saci_0986 (geranylgeranyl reductase), *grsA*: Saci_1585 (GDGT cyclization A), *grsB*: Saci_0240 (GDGT cyclization B), and *Tes*: Saci_0703 (tetraether synthase).

**Table 1 ijms-24-07328-t001:** Ppm error of the lipid species observed in this study.

Lipid Species	Theoretical *m*/*z* ^1^	Observed *m*/*z*	Ppm Error
DOPG	773.533 [M-H]^−^	773.533	0.34
DGGGP	715.507 [M-H]^−^	715.507	−0.08
AA	731.632 [M-H]^−^	731.633	0.56
IP-DGD	893.685 [M-H]^−^	893.685	0.02
CQH	630.451 [M]	630.451	0.22
CQ (6:1)	627.428 [M-H]^−^	627.427	−0.21
CQ (6:2)	625.412 [M-H]^−^	625.412	0.03
CQ (6:3)	623.396 [M-H]^−^	623.396	−0.13
GDGT-0	1346.314 [M + CHO_2_^−^]^−^	1346.310	−2.99
GDGT-1	1344.298	1344.293	−3.33
GDGT-2	1342.283	1342.277	−3.78
GDGT-3	1340.266	1340.263	−3.04
GDGT-4	1338.251	1338.244	−4.75
GDGT-5	1336.235	1336.234	−1.36
GDGT-6	1334.220	1334.219	−0.79
GDGT-7	1332.204	1332.204	−0.32
GDGT-8	1330.188	1330.188	−0.67
DH-GDGT(0)	1670.419 [M + CHO_2_^−^]^−^	1670.412	−4.05
DH-GDGT(1)	1668.404	1668.398	−3.60
DH-GDGT(2)	1666.388	1666.382	−3.44
DH-GDGT(3)	1664.372	1664.368	−2.77
DH-GDGT(4)	1662.357	1662.350	−3.86
DH-GDGT(5)	1660.341	1660.340	−0.61
DH-GDGT(6)	1658.293	1658.288	−3.11
DH-IP-GDGT(0)	1866.433 [M-H]^-^	1866.433	0.17
DH-IP-GDGT(1)	1864.417	1864.410	−3.68
DH-IP-GDGT(2)	1862.402	1862.396	−3.28
DH-IP-GDGT(3)	1860.386	1860.381	−2.75
DH-IP-GDGT(4)	1858.370	1858.369	−0.96
DH-IP-GDGT(5)	1856.355	1856.354	−0.43
DH-IP-GDGT(6)	1854.339	1854.338	−0.54
PH-IP-GDGT(0)	1542.327 [M-H]^−^	1542.321	−4.27
PH-IP-GDGT(1)	1540.312	1540.306	−3.47
PH-IP-GDGT(2)	1538.296	1538.292	−2.82
PH-IP-GDGT(3)	1536.280	1536.277	−2.49
PH-IP-GDGT(4)	1534.265	1534.263	−1.13
PH-IP-GDGT(5)	1532.249	1532.248	−0.72
PH-IP-GDGT(6)	1530.233	1530.233	−0.54

^1^ Lipid species with unspecified mass adducts have the same adducts as the core molecule.

## Data Availability

All the data in this study have been provided in the manuscript, and the raw data are available as part of the Appendix A.

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
