# Peer review of "Membrane Adaptations and Cellular Responses of Sulfolobus acidocaldarius to the Allylamine Terbinafine"

_ijms, 2023, doi:10.3390/ijms24087328_

Round 1

Reviewer 1 Report (Previous Reviewer 1)

Comments for

General Comments: The paper by Rao et al “Membrane adaptations and cellular responses of Sulfolobus acidocaldarius to the allylamine terbinafine” describes a study in which an archaeal organism is treated with an antifungal agent. The presumed action of this agent on S. acidocaldarius is to inhibit lipid synthesis. Given that much remains to be learned about archaeal lipids composition and membranes, this could be a study of value. However, in the present form, the value of this manuscript is low. This appears to be a hastily compiled summary and analysis of the data. For example, the text still contains the tracked changes from a prior editing. In addition, insufficient data is provided to allow a thorough review. The manuscript is completely lacking in statistical analyses. Now overview of the LCMS data (nor the data it self) is provided, so it is not possible to assess the results. The authors need to provide a clear and compelling reason for why the use of terbinafine is relevant to the study of this organism and what biological and biochemical insights come from this.

Specific Comments:

The writing needs to clarified in the abstract and throughout the text. For example, “early cellular response” is used without clarifying what this means. Statement over state the data such as “Transcriptomic data indicated that terbinafine primarily targets the respiratory complex along with genes involved in motility, fatty acid metabolism and GDGT cyclization.” are misleading. There were a wide range of changes in the data, it is more accurate to say that the authors chose to focus of these changes in the data rather than that this is the primary target.

The introduction is difficulty to follow as it works its way through a string of molecules and enzymes. This could be improved with the addition of a figure/scheme such as the one submitted as supplemental material.

The authors have failed to follow standard convention for the requirement of making an identification with LCMS. The brief statement on using an in-house library made from in-silico data does not provide sufficient information.

The authors need to provide a justification for using 5 significant figures beyond the decimal point in the m/z value. Even an orbitrap does not have that accuracy (100,000th of an amu). This also is not consistent with the tolerances used in the searches of 5 ppm. This brings up another point needing clarification – are the m/z values in the text expected or measured values? What was the ppm error in the measurement?

The LCMS data appears to have been significantly manipulated to construct the figures. The traces are described as EICs but are continuous across the chromatogram, however, it is impossible to know when one specific m/z ends and another begins. If chromatographic data is going to be shown, then a TIC must also be shown.

The statistical analysis used to construct Figure 4 should be included as part of the supplemental material. The also needs to be a list of each gene that has been grouped into the categories that are listed in the supplement as well.

The panels showing MS/MS data are of low quality and the data appears to have been manipulated (there is no baseline signal. While cleavage points for the observed fragments have are show, these are not the most obvious bonds to break nor do the account for all of the major ions in the spectrum. Since an orbitrap was used, the authors should be able to confirm the presence of a sulfur atom in the ring structure by showing the two isotopes (32/34) in the isotope pattern for the ions at 208 and 222.

These are only a sampling of the issues with the data and text and should be taken only fas examples rather than all points that the authors need to address.

Author Response

Reviewer 2 Report (New Reviewer)

The manuscript describes the effects of the potent antibiotic terbinafine on the cell membrane of Sulfolobus. The manuscript is well written and accurately designed. The obtained results from the described experiments are of interest and well presented, and the results were thoroughly discussed in the light of previous findings. I recommend acceptance of the manuscript. 

Author Response

We thank the reviewer for their feedback and comments, this has helped in improving the readability of the manuscript.

Round 2

Reviewer 1 Report (Previous Reviewer 1)

The significant figures on the m/z values still do not adhere to the standards of analytical measurement. Retaining non significant figures does not reduce error in calculations of accuracy. It only makes the errors appear smaller than they actually are.

Author Response

We thank the reviewer for their suggestion.

We have now updated the figures, table 1 and the text with the m/z up to 3 significant figures as suggested by the reviewer 1.

This manuscript is a resubmission of an earlier submission. The following is a list of the peer review reports and author responses from that submission.

Round 1

Reviewer 1 Report

Membrane adaptations and cellular responses of Sulfolobus acidocaldarius to the allylamine terbinafine

Overall impression: The manuscript by  Rao and Driessen  sets out to delineate the impact of terbinafine on the archaeal organism Sulfolobus acidocaldarius. Given the paucity of studies focused on lipid synthesis and membrane composition in archaea in general, there is potential value in this work. However, in the current form this manuscript has significant shortcomings. Given the extent of the issues, the points described below are only a partial list of changes needed. 

Major issues

The authors never report at what time cells were collected for exponential and stationary phase analyses. Based on the results in Fig 2 some of the cultures were still in exponential growth phase at 72h.

What is the reason for such a pronounced standard error at the later time points, especially in case of highest terbinafine concentration? While culturing of Sulfolobus is not like E. coli, this raises the question about cell viability. If the error is due to cell death, then this significantly complicates the interpretation of the results.

Experimental design includes three experiments: effect of terbinafine on cell growth, membrane composition and transcription. Unfortunately, each experiment uses a different volume culture (5ml, 10ml and 200ml). Unless grown in a chemostat where chemical and physical parameters can be monitored and adjusted, different volumes of cultures can have a large impact on cell physiology. Even small changes in the rpm of a shaker or the surface to volume ratio of a batch culture can have profound impacts on growth. To conduct this experiment properly, the cells should be grown in the same size flasks and even better, the cells from one flask should be divided between the experiments.

The transcriptomics results are based on two replicates. This means that a robust statistical analysis is not possible. Given the points the authors are trying to make, this is not acceptable.

The most significant shortfall is however related to the lipid species in question. No information is provided on how the lipid species where identified. Lipids in general and challenging to identify with confidence and the issue is more pronounced with archaeal lipids as standards are difficult to obtains and fragmentation patterns are often not found in databases. Without a detailed description of the process used included MS/MS spectra for each of the lipid classes, it is impossible to properly review this work because no basis for confidence in the data is available.

Minor Issues

In general, the grammar is clear however, the writing is often choppy and does not flow from on point to the next. This is particularly obvious in the introduction. Based on the style, it appears the Discussion may have been written by a different author or at a different time.

The rational behind using 0.31mM and 1.2mM concentrations of terbinafine should be explained to the reader.

Line 33-34: Same acronym used twice for two different things, a product and a protein.

lots of similar acronyms here.  Could get complicated.  A mini guide may be useful for all the acronyms. 

Citations jump from 1 to 3.  No citation 2.

Line 45 and 46:  Citation needed.

Line 54 and 55:  This is an awkward sentence.  Why is the second part of the sentence significant in context with the paragraph?  The link is not made clear here.

Line 60-61:  Needs a citation. 

Line 73: This is one of the main observations that prompted the study. Where did the observation come from?

Line 83-84:  Logic behind should be clarifieed

The figure quality is variable. Some are very nice, others are of low quality. Most of them need large font on the axes and labels to be easily readable.

The results are reported in a confusing way. Figure 6 is an example where it seems an incorrect interpretation of the data has been made. In the text we read that CQH is generated ONLY in absence of terbinafine. This is not the case, because detectable levels are present in both exponential and “stationary” phase.  Another issue here is data “normalization”. Based on TIC the dominant form upon inhibitor treatment (in both tested growth phases) is CQ (6:3). However, based on peak area, the dominant form would be CQ (6:1). How is it possible to have a four times larger peak area from a peak that appears to be four times smaller (on the level of e+05 ion count)? 

What is a purpose of adding internal standard to sample prep, if there is no reference to it in the entire manuscript?

In the Methods Section:

a)      section 4.2 misses the inhibitor concentrations used in experiment; what is the meaning of “read were visualized using multiQC” (line 327)?

b)     section 4.3 is missing information on when the internal standard was added; how lipid concentration was estimated (0.25mg/mL); capillary voltage should have negative value (line 348); what is miliQ (line 349)?; what is the ammonium formate pH (line350); autocorrect: line 352 should be “generic”

c)      It was stated that final concentrations of 0.31, 0.61 and 1.22 mM were used, but nowhere in the results do they discuss using a concentration of 0.61mM.

In section 2.5 cellular motility: transcript levels have different elevated/reduced designation than in Fig S4 (lines 207-208); arnR is missing in the figure

Data not shown: If it is not shown then it should not be mentioned in the manuscript.

Incorrent figure called out in text: lines 192 and 194

Throughout the manuscript the authors improperly refer to p-values being adjusted to 0.05 for significance.

Latin species names should be in italic.

Reviewer 2 Report

The presented article is devoted to the biosynthesis of lipids in archaeal membranes of cells (S. acidocaldarius) and the effect of terbinafine (allylamine drug, the squalene epoxidase inhibitor) on the lipid biosynthesis and membrane adaptations.  The research methodology corresponds to that used in the cell biochemistry and the results are obtained for the first time.

The following remarks need to be corrected:

- The advantage of the study is that the authors provide intervals for errors, not just means. However, sometimes this leads to the additional questions. If the reader is not aware that the growth curve should have an exponential growth phase and a saturation phase, then the curve with the maximum dose of terbinafine may seem more like a straight line (Figure 1, 1.22mM). This impression is created because the determination errors for the last two points are too large. The authors need to comment on this.

- The authors write “representative extracted ion chromatograms” in the captions to the figures with chromatograms. It is probably better to write reverse chromatography according to the type of the stationary phase used. Since the chromatogram gives dependence on the retention times which are determined primarily by the phase. The collecting of total ion current determined by MS is only the detection method. One could use absorption or refraction to determine peak areas. It is better to specify arbitrary units as the TIC dimension, rather than normalized ones. The use of normalized areas needs some explanation. What do the authors mean - the ratio of a given area to the sum of the areas of all peaks?